# SparsitySolver: Efficient Reinforcement Learning-based Pruning for LLMs

## Abstract

Large Language Models (LLMs) have achieved significant success in the field of Natural Language Processing (NLP). However, due to their large model size and high inference costs, the application of LLMs is restricted. Pruning is regarded as an effective method to reduce the size of LLMs. Mainstream pruning methods for LLMs typically apply a uniform ratio to prune all the layers or determine layerwise sparsity based on simple criteria. Such manually or semi-manually designed pruning strategies often lead to suboptimal results, which makes reinforcement learning a feasible solution. However, current reinforcement learning-based pruning methods usually have redundant environment designs or multiple agents, rendering them ill-suited to massive LLMs. Hence, we propose SparsitySolver, which first incorporates reinforcement learning into the pruning of LLMs, supporting various pruning granularity. SparsitySolver employs an improved reinforcement learning environment, allowing for a rapid pruning strategy search with a small-scale agent. Moreover, to lessen the performance decline caused by structured pruning, we propose a compensation method capable of restoring performance without introducing additional parameters to the model. We evaluate our approach on LLaMA-V1/V2, Mistral, and the OPT families across multiple pruning granularities, achieving performances surpassing the state-of-the-art methods.

## 1 Introduction

Large Language Models (LLMs) have demonstrated outstanding performance in a wide range of language tasks Zhang et al. (2022); Brown et al. (2020); Bubeck et al. (2023); Touvron et al. (2023a;b). However, LLMs come with a substantial model size and high inference costs, meaning deploying pre-trained models demands expensive computational resources. Hence, techniques aiming at reducing the size and computational demands of LLMs, commonly known as model compression, are gaining increasing attention. Numerous compression methods for LLMs have been introduced, encompassing distillation, quantization, and pruning Hu et al. (2021); Frantar et al. (2022); Xiao et al. (2023); Lin et al. (2023); Lee et al. (2023); Frantar & Alistarh (2023); Ashkboos et al. (2024).

Pruning is an effective method to reduce the quantity of model parameters and computations. Considering the substantial cost of fine-tuning for LLMs, mainstream research concentrates on post-training pruning of LLMs without fine-tuning Frantar & Alistarh (2023); Ashkboos et al. (2024); An et al. (2024); Wang et al. (2024); Sun et al. (2023). In this area, several pruning methods for LLMs, including Wanda Sun et al. (2023), SparseGPT Frantar & Alistarh (2023) and SliceGPT Ashkboos et al. (2024), opt to prune LLMs using uniform sparsity ratios per layer. Compared to the costly global pruning An et al. (2024), such a uniform strategy is simpler and more suitable for large-scale LLMs. In addition, certain approaches propose layerwise sparsity ratios that are non-uniform, an example being OWL Yin et al. (2023) that determines the layerwise sparsity ratio based on the proportion of outliers in each layer, while BESA Xu et al. proposes searching for the optimal pruning rate for each layer in a differentiable manner. However, based on previous studies Wang & Tu (2020); Fang et al. (2023), the differences and inter-dependencies between various layers of the model constitute a complex issue. Factors such as the type of each layer, its location within the network, and the associated operators, all influence the appropriate sparsity ratio for that layer. Thereby, a question arises: *what is the most suitable sparsity strategy for large language models?*

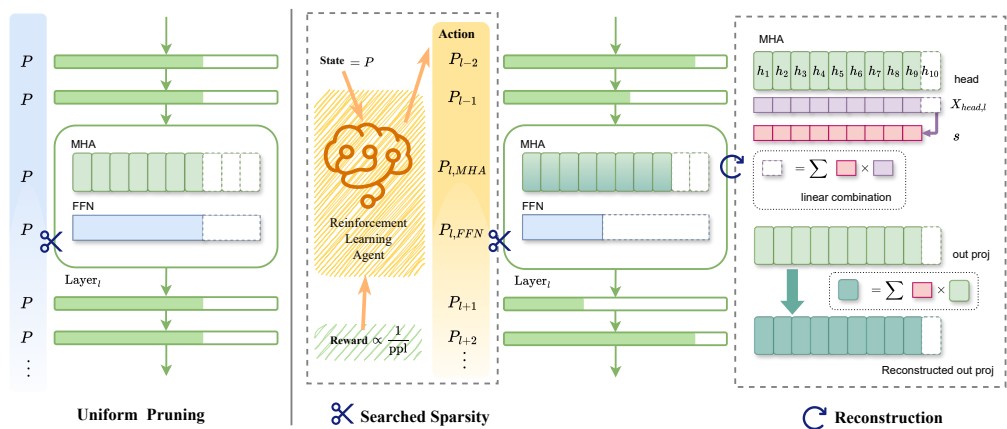

Figure 1: An overview of SparsitySolver. Left: Uniform pruning. Right: SparsitySolver. SparsitySolver first employs a reinforcement learning agent to search for the sparsity ratios for each prunable unit. The agent receives the total pruning ratio from the environment as the state, provides the network's sparsity strategy as an action within one step, and finally, evaluates the perplexity of the pruned network as a reward given to the agent. Second, when using structured pruning, we reconstruct the parameters of the last linear layer of the pruned module as a recovery. Specifically, we compensate for the pruned channels with the linear combination of other channels.

Given that the massive structure of LLMs usually includes dozens of decoder layers, each containing numerous parameters, the search space for pruning strategy is immense. Thus, manually designing or brute-force exploring sparsity strategy becomes nearly impossible. Reinforcement learning poses a solution to this challenge, with various methods exploring the use of reinforcement learning for pruning strategy search He et al. (2018); Alwani et al. (2022); Yu et al. (2021; 2022). However, the environments constructed by these methods are overly redundant, making them unsuitable for massive LLMs. Also, their environment design is skewed. In the environment designed by AMC He et al. (2018), the rewards for intermediate layers pruning are all zero. Only once the pruning of the final layer is done, can a valid reward be assessed on the test set. This type of environment is abnormal as reinforcement learning finds it difficult to manage sparse reward scenarios. Therefore, we suggest simplifying the pruning environment as a solution to correct the sparse reward and avoid additional computations arising from dealing with the environment.

To efficiently and accurately explore the suitable pruning strategy for LLMs, we propose SparsitySolver, a reinforcement learning-based post-training pruning method for LLMs, supporting both structured and unstructured pruning. Furthermore, to address the model damage after structured pruning, we introduce a compensation method to recover the model's performance. Fig. 1 illustrates an overview of our approach. The contributions are summarized as follows:

- We propose a simple and efficient reinforcement learning environment, improving the sparse reward environment in existing RL pruning methods without the need for additional computation. Within our developed environment, a small-scale RL agent is enough to attain quick convergence, thereby making it apt for searching pruning strategies for LLMs.

- To mitigate the performance loss after structured pruning, we propose reconstruction compensation, which requires no additional parameters for recovery.

- We conduct multiple experiments on LLMs including OPT Zhang et al. (2022), Mistral Jiang et al. (2023), LLaMA-V1 Touvron et al. (2023a), and LLaMA-V2 Touvron et al. (2023b) families, verifying that SparsitySolver can explore more suitable sparsity strategies in both structured and unstructured pruning, demonstrating better perplexity than corresponding state-of-the-art methods.

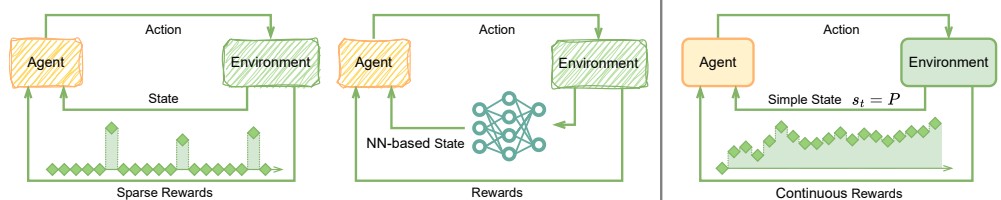

Figure 2: The comparison of reinforcement learning environments. Left: Environment with sparse rewards, like AMC He et al. (2018). Middle: Environment that processes states through a neural network, like AGMC Yu et al. (2021). Right: The environment we proposed, featuring simple states and continuous rewards.

## 2 RELATED WORKS

**LLM Pruning** Many studies Frantar & Alistarh (2023); Sun et al. (2023); Yin et al. (2023); Ashkboos et al. (2024); Ma et al. (2023); An et al. (2024) indicate that pruning is a practical approach for reducing the scale of LLMs. SparseGPT, according to Frantar & Alistarh (2023), utilizes the inverse of the Hessian matrix to prune and subsequently update weights. In a different study, Wanda Sun et al. (2023) brings a new pruning criterion for LLMs, combining the weight magnitude and input activations to retain outliers. Carrying this forward, OWL Yin et al. (2023) introduces a non-uniform layerwise sparsity ratio, decided based on the proportion of outliers in each layer. BESA Xu et al. proposes searching for the optimal pruning rate for each layer in a differentiable manner. However, the above methods either only support or mainly support unstructured pruning. Since unstructured pruning leads to irregular sparse patterns and requires specialized hardware support, other approaches are centered around the exploration of structured pruning. LLM-Pruner Ma et al. (2023) performs structured pruning based on gradient information, and then conducts fine-tuning using LoRA Hu et al. (2021). SliceGPT Ashkboos et al. (2024) converts the LayerNorm into RMSNorm, performs a transformation on every block of the model through computational invariance, and then carries out the corresponding pruning. FLAP An et al. (2024) proposes a framework that includes global structure search and baseline bias compensation. However, the above methods either require fine-tuning or introduce additional parameters. For example, the baseline bias compensation in FLAP introduces bias to linear layers that are initially without bias. The transformation in SliceGPT goes even further by adding a new linear layer to each skip connection. These compensation methods, which require the introduction of additional parameters, contradict the original intent of pruning and may impact subsequent deployment and inference.

**Reinforcement Learning-based Pruning** Several methods He et al. (2018); Alwani et al. (2022); Chen et al. (2020); Yu et al. (2021; 2022) propose utilizing RL agents to search for pruning strategies. AMC He et al. (2018) first suggests using reinforcement learning agents to explore pruning strategies. Chen et al. (2020) propose a deep reinforcement learning-based runtime pruning method, where a runtime agent and a static agent jointly make sparsities. DECORE Alwani et al. (2022) utilizes multi-agent reinforcement learning to determine whether each channel should be pruned. GNN-RL Yu et al. (2022) and AGMC Yu et al. (2021) employ Graph Neural Networks to capture the features of the pruned network, and then use reinforcement learning to search for effective pruning strategies. However, the aforementioned methods either require complex environment handling, such as GNN-RL and AGMC, need cooperation among multiple agents, like in the case of DECORE and DRL-based methods, or involve abnormal environments like AMC. Such complex methods are unsuitable for exploring pruning strategies in large-scale LLMs.

## 3 METHODOLOGY

### 3.1 EXPLORING WITH REINFORCEMENT LEARNING

A number of approaches suggest using reinforcement learning agents to search for pruning strategies within Convolutional Neural Networks He et al. (2018); Alwani et al. (2022); Yu et al. (2021;

2022). However, the reinforcement learning environments constructed by these methods tend to be over-complicated and ill-suited for large-scale LLMs. As shown in Fig. 2, we propose simplifying the pruning environment as a solution to rectify sparse rewards and avoid additional computations incurred by processing the environment. In the following, we provide a detailed description of the reinforcement learning setup.

**State Space**    In our proposed pruning environment, we define the state as:

$$S_t = P \tag{1}$$

where $P$ is the total pruning ratio. Such a design of the state space not only negates the need for extra computations but also eliminates the necessity for dynamic implicit modeling of the environment. At this point, the agent can be regarded as a differential mapping of the pruning strategy to rewards, which means the agent directly models the actions. Experiments show that the simplified state space does not affect the performance of reinforcement learning. On the contrary, such a simplified state design accelerates the speed of the search, with specific details provided in Sec. 4.6.

**Action Space**    The action given by the RL agent is the preserved ratio for every layer within a continuous space, which is defined as:

$$A_t = [a_1, a_2, \cdots, a_N] \in \mathbb{R}^N \tag{2}$$

where $a_i \in [a_{min}, a_{max}]$, $a_{min}$ and $a_{max}$ are the lower and upper bounds on the sparsity rate for each layer. $N$ represents the number of prunable units in the network. In the case of structured pruning, prunable units refer to Multi-Head Attention (MHA) layers and Feed-Forward Network (FFN) layers. For MHA layers, we carry out pruning at the granularity of attention heads. In unstructured pruning, prunable units are defined as weight matrices. For the agent-given action $A_t$, we

---

**Algorithm 1** Action Constraints

**Initial:** The number of parameters per prunable unit $W = [w_1, w_2, \cdots, w_N]$, total number of parameters $W_{all}$, lower bound $a_{min}$ and upper bound $a_{max}$.
**Input:** The original action $A_t = [a_1, a_2, \cdots, a_N]$ provided by the agent and the total pruning ratio $P$.
**Output:** The constrained action $\tilde{A}_t = [\tilde{a}_1, \tilde{a}_2, \cdots, \tilde{a}_N]$.
  1: $A_t \leftarrow \tanh(A_t + 1)/2$
  2: $A_t \leftarrow A_t \times (a_{max} - a_{min} + 0.1) + a_{min}$
  3: $A_t \leftarrow \text{clip}(A_t, 0, a_{max})$
  4: **for all** $a_i$ in $A_t$ **do**
  5:     $a_i \leftarrow \text{Round}(a_i \times w_i)/w_i$
  6: **end for**
  7: **for all** $a_i$ in $A_t$ **do**
  8:     $W_{other} \leftarrow \sum_{k<i} \tilde{a}_k \times w_k + \sum_{k>i} a_{min} \times w_k$
  9:     $\tilde{a}_i = \min(a_i, ((1 - P) \times W_{all} - W_{other})/w_i)$
 10: **end for**

---

need to enforce constraints on it to obtain $\tilde{A}$, as illustrated in Alg. 1. After obtaining the sparsity strategy $\tilde{A}$, we prune the network using the derived strategy.

For the pruning criteria, our method is compatible with most of the mainstream pruning criteria currently in use. In unstructured pruning, we choose Wanda Sun et al. (2023) as the pruning criterion, while in structured pruning, we opt for the $\ell 2$-norm of the activations as the criterion. It is worth noting that our layer-by-layer pruning does not require any global information or gradient information as a pruning criterion, which is memory-friendly.

**Reward Function**    Given the above-mentioned state space and action space, the policy only needs to execute one step per episode. After pruning the model with the searched strategy, we obtain a model that meets the total pruning ratio $P$ and subsequently evaluate the pruned model according to the task metric. Considering that our experiments are primarily performed on WikiText Merity et al. (2016) and perplexity is used as the evaluation metric, we define the default reward function as $R = \frac{10}{ppl}$, where $ppl$ is the perplexity evaluated on the WikiText validation. We expect the final convergence value to fall within the range of $(1, 2)$, remaining within the same order of magnitude. Based on current LLM benchmarks, we set the coefficient of the reward function to 10.

**Proximal Policy Optimization (PPO)**    Multiple reinforcement learning algorithms aim to search within continuous action spaces, examples include Deep Deterministic Policy Gradient (DDPG) Lillicrap et al. (2016), Proximal Policy Optimization (PPO) Schulman et al. (2017), and Soft Actor-Critic (SAC) Haarnoja et al. (2018). We utilize PPO as the reinforcement learning algorithm for search due to its highly efficient policy. Essentially, we only require our agent to learn a differentiable mapping from pruning strategy to rewards. Given the simplicity of our designed environment state, we can further reduce the size of the critic network.

## 3.2 COMPENSATION THROUGH RECONSTRUCTION

Our method supports both structured and unstructured pruning. To reduce the negative impact on network performance caused by structured pruning, we propose compensating for the pruned MHA layers and FFN layers. Inspired by the data-free compression method UDFC Bai et al. (2023), we assume that the channels that are damaged due to pruning can be restored through a linear combination of other channels.

For each pruned module, we consider the last linear layer $W^{out}_{:,:} \in \mathbb{R}^{N_{out} \times N_{in}}$ as the reconstruction layer. For the pruned channel $W^{out}_{:,p}$, the assumption of recovery can be formulated as follows.

$$W^{out}_{:,p} \approx \sum_{j=1, j\notin\mathcal{P}}^{N_{in}} s_{p,j} \times W^{out}_{:,j}, \quad \forall p \in \mathcal{P} \tag{3}$$

where $s_{p,j}$ is a scale factor that evaluates the level of association between the $j$-$th$ channel and the $p$-$th$ channel, and $\mathcal{P}$ is the set of indices of the pruned channels. At this point, the preserved channel $W^{out}_{:,j}$ can be written as:

$$W^{out}_{:,j} = W^{out}_{:,j} + \sum_{p\in\mathcal{P}} s_{p,j} \times W^{out}_{:,p}, \quad \forall j \in [1, N_{in}], \quad j \notin \mathcal{P} \tag{4}$$

Without pruning, the $k$-$th$ output channel of the last linear layer can be represented as:

$$X^{(\ell+1)}_k = \sum_{j=1}^{N_{in}} W^{out}_{k,j} X^{(\ell)}_j + b^{out}_k \quad \forall k \in [1, N_{out}] \tag{5}$$

where $X^{(\ell)}_j$ represents the input corresponding to each input channel $W^{out}_{k,j}$ and $b^{out}_k$ represents the bias, which can be 0. Without loss of generality, we assume that only the $p$-$th$ input channel is pruned. After pruning and reconstruction with equation 4, the $k$-$th$ output channel can be represented as follows.

$$\hat{X}^{(\ell+1)}_k = \sum_{j=1, j\neq p}^{N_{in}} (W^{out}_{k,j} + s_{p,j} \times W^{out}_{k,p})X^{(\ell)}_j + b^{out}_k \quad \forall k \in [1, N_{out}] \tag{6}$$

The reconstruction error $\ell_{re}$ of the $k$-$th$ output channel can be defined as:

$$\ell_{re} = \|X^{(\ell+1)}_k - \hat{X}^{(\ell+1)}_k\|_2^2 = \|W^{out}_{k,p} X^{(\ell)}_p - \sum_{j=1, j\neq p}^{N_{in}} s_{p,j} \times W^{out}_{k,p} X^{(\ell)}_j\|_2^2$$

$$= \|W^{out}_{k,p}(X^{(\ell)}_p - \sum_{j=1, j\neq p}^{N_{in}} s_{p,j} \times X^{(\ell)}_j)\|_2^2 \tag{7}$$

Note that pruning does not change $W^{out}_{k,p}$. Therefore, we further define the reconstruction error as:

$$\ell_{re} = \|X^{(\ell)}_p - \sum_{j=1, j\neq p}^{N_{in}} s_{p,j} \times X^{(\ell)}_j\|_2^2 + \lambda \sum_{j=1, j\neq p}^{N_{in}} \|s_{p,j}\|_2^2 \tag{8}$$

where $\lambda$ is a non-negative penalty coefficient. Next, by minimizing the reconstruction error, we prove the existence of the optimal solution $s$. For simplicity, we define:

$$\mathbf{X}_p = [X^{(\ell)}_p], \quad \mathbf{X} = [X^{(\ell)}_j], \quad \mathbf{s} = [s_{p,j}]. \qquad j \in [1, N_{in}], \quad j \neq p \tag{9}$$

The reconstruction error can be simplified as $\ell_{re} = (\mathbf{X}_p - \mathbf{s}\mathbf{X})^\top (\mathbf{X}_p - \mathbf{s}\mathbf{X}) + \lambda \mathbf{s}^\top \mathbf{s}$. The first and second derivative of the $\mathbf{s}$ is:

$$\frac{\partial \ell_{re}}{\partial \mathbf{s}} = -2\mathbf{X}^\top \mathbf{X}_p + 2\mathbf{s}(\mathbf{X}^\top \mathbf{X} + \lambda \mathbf{I}), \quad \frac{\partial^2 \ell_{re}}{\partial^2 \mathbf{s}} = 2\mathbf{X}^\top \mathbf{X} + 2\lambda \mathbf{I} \tag{10}$$

It can be seen that $\ell_{re}$ is a convex function and there exists a unique optimal solution $s$ such that $\frac{\partial \ell_{re}}{\partial \mathbf{s}} = 0$.

We generalize the $p$-$th$ pruned channel to all pruned channels set $\mathcal{P}$. For each pruned channel, we can solve equation 8 through Ridge regression to obtain a set of scale factors $s_{p,:}$. With these scale factors, we can recover the pruned channels using a linear combination of the preserved channels, as shown in equation 4. Notably, our reconstruction method only updates the weights of the last linear layer and avoids introducing any extra parameters.

## 4 EXPERIMENTS

**Models and Datasets**   We evaluate the performance of SparsitySolver across a series of LLMs, including the OPT Zhang et al. (2022), LLaMA-V1 Touvron et al. (2023a), LLaMA-V2 Touvron et al. (2023b) and Mistral Jiang et al. (2023) model families. Our evaluation aligns with the existing LLM pruning methods Ashkboos et al. (2024); An et al. (2024); Yin et al. (2023), including perplexity assessment on the WikiText Merity et al. (2016) validation and evaluation on seven common-sense benchmarks (BoolQ Wang et al. (2019), OpenbookQA Mihaylov et al. (2018), WinoGrande Sakaguchi et al. (2021), HellaSwag Zellers et al. (2019), PIQA Bisk et al. (2020), ARC-e, and ARC-c Clark et al. (2018)) in zero-shot setting consistent with the LM-Evaluation-Harness Gao et al. (2021).

**Baseline**   We select the corresponding LLM pruning baseline according to specific pruning granularities. In terms of unstructured pruning, we employ Wanda Sun et al. (2023), BESA Xu et al., and OWL Yin et al. (2023) as comparative baselines. In terms of structured pruning, we compare SparsitySolver with the SOTA post-training structured pruning methods SliceGPT Ashkboos et al. (2024) and FLAP An et al. (2024). We also compare the singular value decomposition-based method SVD-LLM Wang et al. (2024).

**Setup**   We utilize the Proximal Policy Optimization (PPO) as our reinforcement learning agent. Our actor network comprises two hidden layers, with each layer holding 256 neurons, our critic network also consists of two hidden layers, each of which has 64 neurons. The learning rate is 5e-4, the number of samples per update is 15, the learning epoch is 10, and 3000 episodes are searched.

### 4.1 LANGUAGE MODELING PERFORMANCE

Table 1: WikiText validation perplexity of various unstructured pruning methods on LLaMA-V1-7B/13B at 50%, 60% and 70% sparsity. * indicates the experimental results we reproduced using the open-source code. The **bolded** results indicate the best performance.

| Method | Layerwise Sparsity | Searched Sparsity | 50% | | 60% | | 70% | |
|---|---|---|---|---|---|---|---|---|
| | | | 7B | 13B | 7B | 13B | 7B | 13B |
| Dense | - | - | 5.68 | 5.09 | 5.68 | 5.09 | 5.68 | 5.09 |
| Wanda * | ✗ | ✗ | 7.26 | 6.17 | 10.86 | 8.91 | 93.07 | 57.70 |
| OWL *w.t.* Wanda * | ✓ | ✗ | 7.22 | 6.06 | 9.52 | 7.62 | 24.96 | 17.37 |
| BESA * | ✓ | ✗ | 7.12 | 6.13 | 12.66 | 9.87 | 84.70 | 54.40 |
| Ours | ✓ | ✓ | **6.94** | **6.02** | **9.50** | **7.41** | **22.84** | **16.57** |

Table 2: WikiText validation perplexity of structured pruning methods for OPT-125M/1.3B/2.7B/6.7B/13B, LLaMA-V2-7B/13B/70B and Mistral-7B at 20% sparsity. The dash '-' represents results that could not be reproduced with the open-source code.

| Method | Searched Sparsity | Weight Update | Additional Parameters | OPT | | | | | LLaMA-V2 | | | Mistral |
|---|---|---|---|---|---|---|---|---|---|---|---|---|
| | | | | 125M | 1.3B | 2.7B | 6.7B | 13B | 7B | 13B | 70B | 7B |
| Dense | - | - | - | 27.64 | 14.61 | 12.46 | 10.85 | 10.12 | 5.47 | 4.88 | 3.32 | 5.25 |
| SliceGPT * | ✗ | ✓ | ✓ | 34.10 | 16.51 | 13.89 | 11.60 | 10.71 | 6.84 | 6.06 | 4.25 | 6.96 |
| FLAP * | ✗ | ✓ | ✓ | 34.45 | 17.37 | 15.38 | 12.79 | 13.17 | 7.15 | 6.31 | 4.12 | 6.57 |
| SVD-LLM * | ✗ | ✓ | ✗ | 38.86 | 17.82 | 15.22 | 12.06 | - | 8.38 | 6.66 | 4.66 | - |
| Ours | ✓ | ✗ | ✗ | 31.44 | 17.26 | 14.55 | 13.07 | 11.80 | 7.54 | 6.45 | 4.32 | 6.89 |
| Ours (Recon) | ✓ | ✓ | ✗ | **30.67** | **15.82** | **13.75** | **10.47** | **10.23** | **6.79** | **6.01** | **4.06** | **6.48** |

Table 1 presents the performance of various unstructured pruning methods on language modeling with sparsity levels of 50%, 60%, and 70% on WikiText. Our method achieves a high sparsity rate of 70% in unstructured pruning and outperforms Wanda, OWL, and BESA across a range of sparsity levels. This comparison indicates that the pruning strategy provided by the reinforcement learning agent is superior to the hierarchical layerwise sparsity rates determined by the proportion of outliers in OWL and also outperforms the layerwise sparsity rates searched in a differentiable

manner in BESA, as the RL agent demonstrates stronger capabilities in managing inter-layer differences. More detailed comparisons of unstructured pruning strategies are presented in App. C.

Table 2 presents the performance of various structured pruning methods for LLMs under a sparsity ratio of 20% on WikiText. It can be observed from Table 2 that even if we only use the searched strategy without incorporating reconstruction compensation, our method can still achieve an acceptably low level of perplexity. In the case of certain models (such as OPT-125M), our pure searched strategy achieves better results than other pruning methods incorporating compensation. Furthermore, when the reconstruction compensation is combined with the strategy we searched, our method surpasses other structured pruning techniques, as shown in Table 2 under 'Ours (Recon)'. Figure 3 illustrates a comparison of the PPL-Sparsity pattern Pareto curves for various pruning on the OPT-125M model, showing that the Pareto curve of SparsitySolver with reconstruction compensation significantly outperforms other methods.

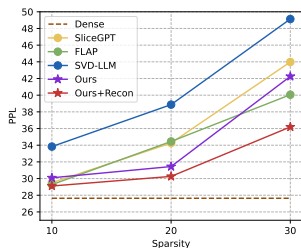

Figure 3: Comparison of the PPL-Sparsity pattern Pareto curve of the OPT-125M.

## 4.2 ZERO-SHOT TASKS PERFORMANCE

Table 3: Zero-shot performance of structured pruning on LLaMA-V1-7B at 20% sparsity. The underlined results indicate the second-best performance.

| Method | Searched Sparsity | Weight Update | Additional Parameters | BoolQ | PIQA | HellaSwag | WinoGrande | ARC-e | ARC-c | OBQA | Ave. |
|---|---|---|---|---|---|---|---|---|---|---|---|
| Dense | - | - | - | 75.02 | 79.16 | 76.20 | 70.09 | 72.85 | 44.62 | 44.40 | 66.05 |
| SliceGPT * | ✗ | ✓ | ✓ | 58.99 | 69.86 | 59.45 | **68.43** | 62.37 | 36.60 | 37.40 | 56.15 |
| FLAP * | ✗ | ✓ | ✓ | 71.44 | 75.14 | 67.71 | 67.40 | 67.21 | 37.46 | **42.00** | 61.19 |
| Ours(C4) | ✓ | ✓ | ✗ | 72.75 | **75.73** | **69.34** | 68.27 | 66.53 | 36.59 | 39.14 | 61.19 |
| Ours(Recon) | ✓ | ✓ | ✗ | **74.37** | 75.29 | 68.41 | 67.85 | **70.23** | **37.47** | 39.60 | **61.89** |

Table 4: Zero-shot performance of unstructured pruning on LLaMA-V1-7B at 70% sparsity.

| Method | Layerwise Sparsity | Searched Sparsity | BoolQ | PIQA | HellaSwag | WinoGrande | ARC-e | ARC-c | OBQA | Ave. |
|---|---|---|---|---|---|---|---|---|---|---|
| Dense | - | - | 75.02 | 79.16 | 76.20 | 70.09 | 72.85 | 44.62 | 44.40 | 66.05 |
| Wanda | ✗ | ✗ | 55.11 | 57.18 | 31.83 | 51.38 | 34.22 | 19.80 | 26.00 | 39.36 |
| OWL * | ✓ | ✗ | 63.48 | 64.90 | 44.79 | 58.72 | 45.03 | 26.19 | 29.60 | 47.53 |
| BESA * | ✓ | ✗ | 57.86 | 55.88 | 31.27 | 50.75 | 32.07 | 22.10 | 27.60 | 39.64 |
| Ours | ✓ | ✓ | **63.97** | **65.67** | **45.16** | **59.22** | **47.35** | **27.13** | **31.80** | **48.61** |

We evaluated the zero-shot capability of our method under structured and unstructured pruning settings across seven downstream tasks, as shown in Tables 3 and 4, respectively. In structured pruning, our model searches for sparsity strategies on WikiText. 'Ours(C4)' represents the compensation results using samples from the C4 Raffel et al. (2020) training set as the calibration set, while 'Ours(Recon)' indicates the compensation results using samples from the respective downstream task dataset as the calibration set. The calibration set consists of 32 samples, each containing 2048 tokens. It is worth noting that our method shows an increase in accuracy in most of the tested downstream tasks, achieving better average results than other methods. Compared to the SliceGPT method, which introduces additional parameters, the total number of model parameters obtained using SparsitySolver is less than that of SliceGPT. When the sparsity rate of the LLaMA-V1-7B model is 20%, the number of parameters after pruning with the SliceGPT method is 6.1B, whereas our method results in only 5.4B parameters, while achieving comparable performance and better average accuracy.

In unstructured pruning, we conducted a direct search on the downstream target dataset, with the reward function defined by zero-shot accuracy. It can be observed that our method outperforms Wanda, OWL, and BESA even at a high sparsity rate of 70%.

## 4.3 SEARCHED STRATEGY

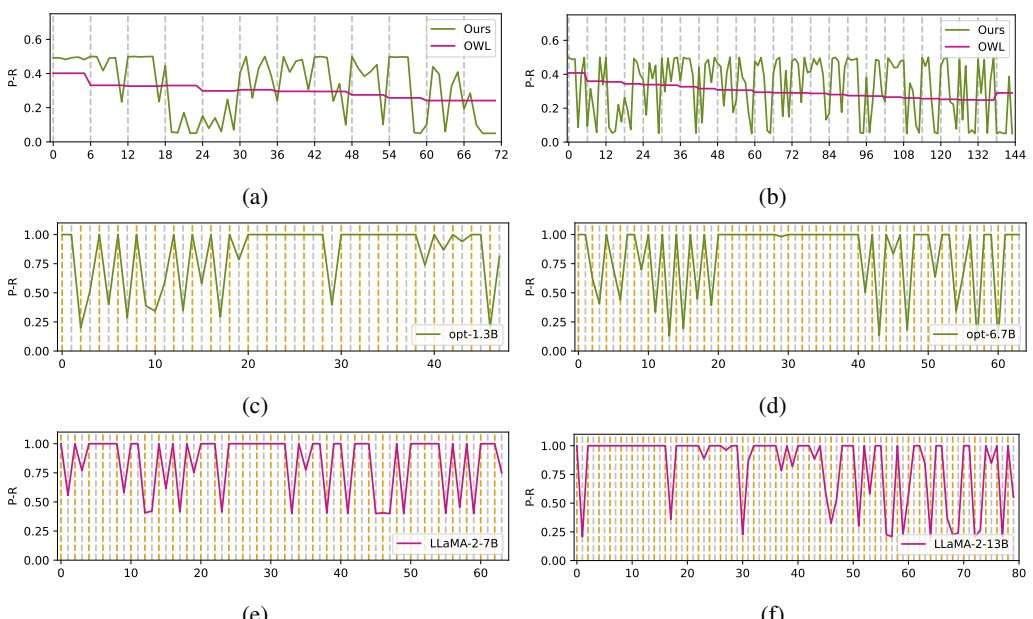

(a)                 (b)

(c)                 (d)

(e)                 (f)

Figure 4: The searched pruning strategies. The horizontal axis is the index of the layer, while the vertical axis "P-R" represents the preserved ratio of parameters in each layer or each weight. (a)-(b): A comparison of unstructured sparsity strategies for OPT-125M and OPT-1.3B under 70% sparsity on WikiText. (c)-(f): The searched pruning strategies of OPT-1.3B/6.7B and LLaMA-V2-7B/13B at a sparsity of 20% under structured pruning. The yellow dashed line represents the MHA layer, and the gray dashed line represents the FFN layer.

We also focus on the pruning strategies obtained from the search. Fig. 4 depicts the pruning strategies obtained from specific models in the OPT and LLaMA families, under the environment and agent we designed. More detailed searched pruning strategies are presented in App. C. Fig. 4a and Fig. 4b show the comparison of the strategies we obtained from the search and OWL's strategies under unstructured pruning. It can be observed that our method is more flexible, allowing different sparsity rates to be assigned to weight matrices at different locations. Fig. 4c and Fig. 4d depict the structured pruning strategies obtained for the OPT series at a sparsity ratio of 20%, with the corresponding perplexities shown in Table 2. Differing from the searched strategies for convolutional neural networks obtained in AMC He et al. (2018), the agent tends to retain more parameters in the middle layers of LLMs and focuses on pruning at the front and back ends of the model. Fig. 4e and Fig. 4f each present the searched strategies of the LLaMA-V2-7B and LLaMA-V2-13B models, respectively. In the LLaMA model, the pruning strategy becomes increasingly complex, making it difficult to summarize a unified trend.

## 4.4 COMPENSATION

For the reconstruction compensation part, we set the default calibration set for compensation to contain 32 samples sampled from the training set, each containing 2048 tokens. The Ridge regression hyperparameter $\lambda$ is set to 0.9. Fig. 5 shows the effect on the compensation results when changing the number of reconstruction samples in the MHA and FFN layers. The results clearly show that for both the MHA and FFN layers, as the number of reconstruction samples increases, the performance improves.

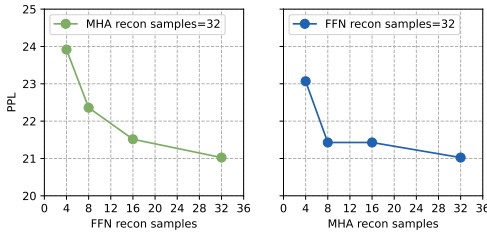

Figure 5: The perplexity of OPT-1.3B at 30% sparsity with different numbers of reconstruction samples used in the MHA and FFN layers.

The GPU time required for reconstruction is related to the sparsity strategy searched. As the searched sparsity ratios vary, the speed of reconstruction also changes. The smaller the sparsity of a layer, the faster its reconstruction speed. On an NVIDIA 80G A800, reconstructing an MHA layer of LLaMA-V2-70B with a 20% sparsity using 32 samples requires 0.21 GPU hours, while reconstructing a 20% sparsity MHA layer of LLaMA-V2-7B and LLaMA-V2-13B takes less than 2 minutes.

## 4.5 ABLATION STUDY

Table 5: A comparison of the pruning results on various models at a 20% sparsity using the different components we proposed. 'Recon' refers to the results obtained using a uniform sparsity strategy combined with the reconstruction, 'Search' indicates the results from pruning with the searched sparsity strategy without reconstruction, and 'Search + Recon' represents the results obtained after applying our search strategy followed by reconstruction.

| Method | Searched Sparsity | Weight Update | OPT | | | LLaMA-V2 | Mistral |
|---|---|---|---|---|---|---|---|
| | | | 125M | 1.3B | 2.7B | 7B | 7B |
| Dense | - | - | 27.64 | 14.61 | 12.46 | 5.47 | 5.25 |
| SliceGPT * | ✗ | ✓ | 34.10 | 16.51 | 13.89 | 6.84 | 6.96 |
| FLAP * | ✗ | ✓ | 34.45 | 17.37 | 15.38 | 7.15 | 6.57 |
| Recon | ✗ | ✓ | 32.25 | 17.15 | 14.65 | 7.14 | 7.22 |
| Search | ✓ | ✗ | 31.44 | 17.26 | 14.55 | 7.54 | 6.89 |
| Search + Recon | ✓ | ✓ | **30.67** | **15.82** | **13.75** | **6.79** | **6.48** |

Additionally, we analyze the effectiveness of each component we proposed, as shown in Table 5, where we conducted ablation experiments on the search strategy and reconstruction. It is evident that both search and reconstruction effectively improve pruning results. In some certain models, both the search-only and reconstruction-only methods yield better results than SliceGPT or FLAP. However, 'Search + Recon', which combines both methods, achieve the highest performance.

## 4.6 REINFORCEMENT LEARNING

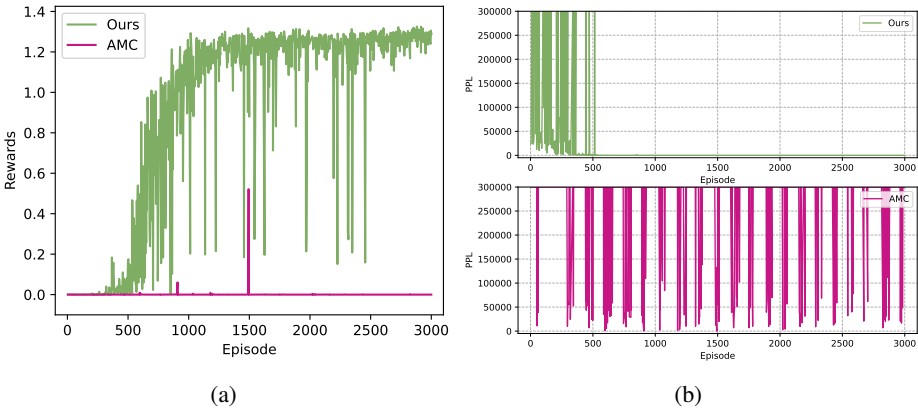

(a)  (b)

Figure 6: Comparison of our proposed reinforcement learning method with the AMC, conducting structured pruning strategy search on LLaMA-V2-7B with a sparsity ratio of 20%. (a) Comparison of the rewards curves. (b) Comparison of the perplexity curves.

We focus on analyzing the performance of our proposed reinforcement learning method. Fig. 6 shows the comparison of our proposed RL agent with AMC He et al. (2018). It is demonstrated that our improved reinforcement learning environment drastically enhances exploration performance. Our continuous reward environment accelerates the convergence speed of the agent, with rewards steadily increasing after 500 episodes. However, it is difficult for the agent to converge when employing AMC's sparse reward environment. As shown by the violet-red curve in Fig. 6b, its perplexity also experiences strong oscillations and remains high. From the reward curve in Fig. 6a,

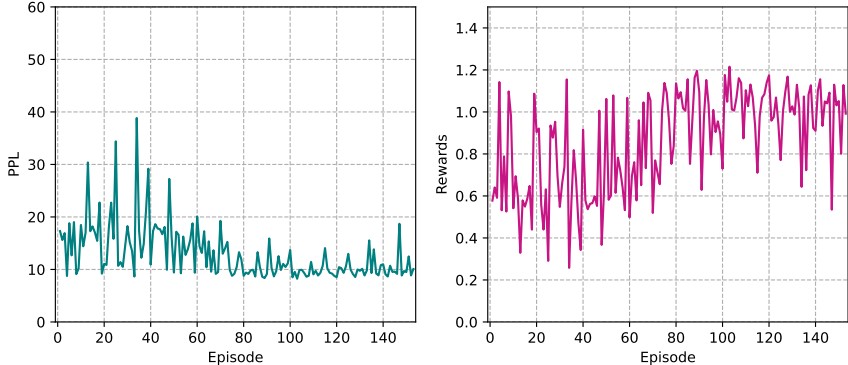

Figure 7: The PPL curve and reward curve for the Mistral-7B model during a 30% sparsity search by the RL agent, which was trained over 1500 episodes at a 20% sparsity rate. Left: The PPL curve. Right: The rewards curve.

it can be seen that our agent achieves convergence between 1500-2000 episodes, delivering a relatively satisfactory outcome. The reinforcement learning curves for other models are provided in App. B, display the same trend. This indicates that a search with 3000 episodes is saturated, and the reinforcement learning design of SparsitySolver is highly efficient, exceeding our expectations.

Moreover, the RL agent trained at one sparsity is reusable for another. We utilized an RL agent trained over 1500 episodes at a 20% sparsity rate to perform a search at a 30% sparsity rate. Figure 7 illustrates the PPL and reward curves during the search process. It can be observed that we achieved relatively good results at around 100 episodes. Table 6 presents the 30% sparsity pruning results obtained by reusing the RL agent trained over 1500 episodes at a 20% sparsity on the OPT-125M/1.3B/2.7B and Mistral-7B models. The 'Episode' indicates the number of episodes required for re-searching and 'Episode time' indicates the time to search for one episode. On two 4090 GPUs, the total search time for Mistral-7B is within 2 GPU hours, and the prun-

Table 6: The pruning results for the OPT-125M/1.3B/2.7B and Mistral-7B models using the reused RL agent at a 30% sparsity.

| Method | OPT | | | Mistral |
|---|---|---|---|---|
| | 125M | 1.3B | 2.7B | 7B |
| Dense | 27.64 | 14.61 | 12.46 | 5.25 |
| SliceGPT * | 44.23 | 19.58 | 16.31 | 9.46 |
| FLAP * | 40.05 | 20.77 | 18.31 | 8.90 |
| Episode | 112 | 135 | 102 | 101 |
| Episode time (s) | 12.62 | 25.86 | 35.06 | 74.48 |
| Ours | 39.53 | 21.71 | 17.51 | 8.22 |
| Ours (Recon) | **36.19** | **18.65** | **16.23** | **7.79** |

ing results for pure search have outperformed SliceGPT's and FLAP's. The trade-off in pruning time for our method is acceptable in practical applications.

## 5 CONCLUSION

We propose SparsitySolver, a reinforcement learning-based method for large language model pruning that allows for various levels of granularity. In order to more effectively explore suitable pruning strategies for LLMs, we introduce reinforcement learning search into LLM pruning for the first time. Through our enhanced reinforcement learning environment, the agent can converge in a short period and quickly derive a pruning strategy. Furthermore, we propose a reconstruction compensation method for structured pruning to recover the model performance without introducing additional parameters. Our experimental results have confirmed the efficacy of our method. We hope our work can provide assistance in the design of future LLM pruning strategies.

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

**Appendix**

## A EXPERIMENTAL DETAILS

Table 7: Parameters for the PPO agent.

| Parameters | Value |
|---|---|
| Actor learning rate | 5e-4 |
| Critic learning rate | 5e-4 |
| Actor hidden size | 256 |
| Critic hidden size | 64 |
| Optimizer | Adam |
| number of samples per update | 15 |
| number of learning epochs | 10 |
| number of episodes | 3000 |

The specific parameters of the PPO agent utilized in the SparsitySolver method are summarized in Table 7. For unstructured pruning, we use Wanda as the pruning criterion, with the number of calibration samples set to 128. For structured pruning, we use the $\ell 2$-norm of the activation values as the pruning criterion, with the number of calibration samples set to 64. During reconstruction compensation, we set the number of reconstruction samples to 32, and $\lambda$ to 0.9. Each sample contains 2048 tokens.

## B REINFORCEMENT LEARNING

Fig. 8 - 10 show the reward curves and perplexity curves for various models during the process of reinforcement learning search.

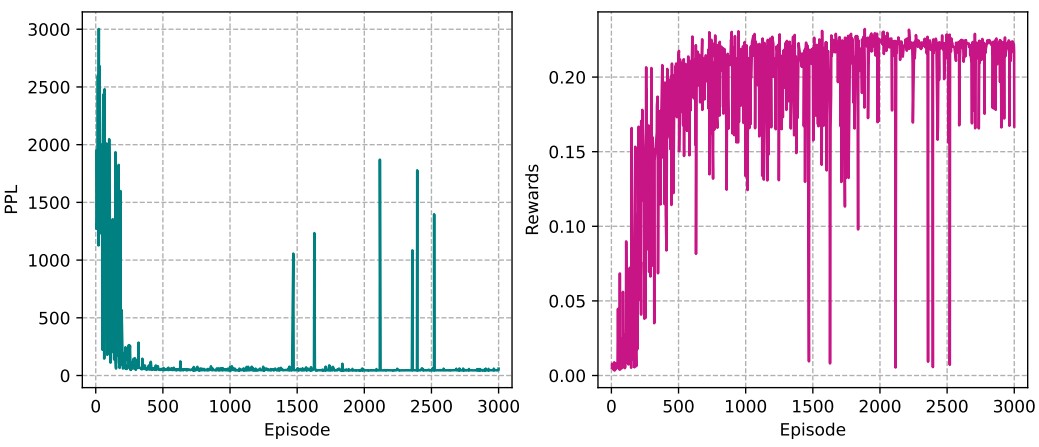

Figure 8: Reinforcement learning reward curve and perplexity curve for OPT-125M under structured pruning with a sparsity ratio of 20% on Wikitext.

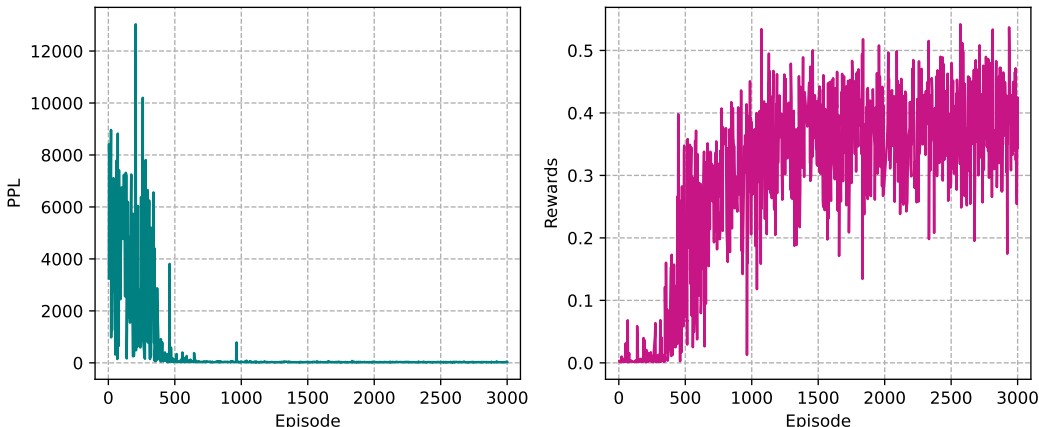

Figure 9: Reinforcement learning reward curve and perplexity curve for OPT-1.3B under structured pruning with a sparsity ratio of 20% on Wikitext.

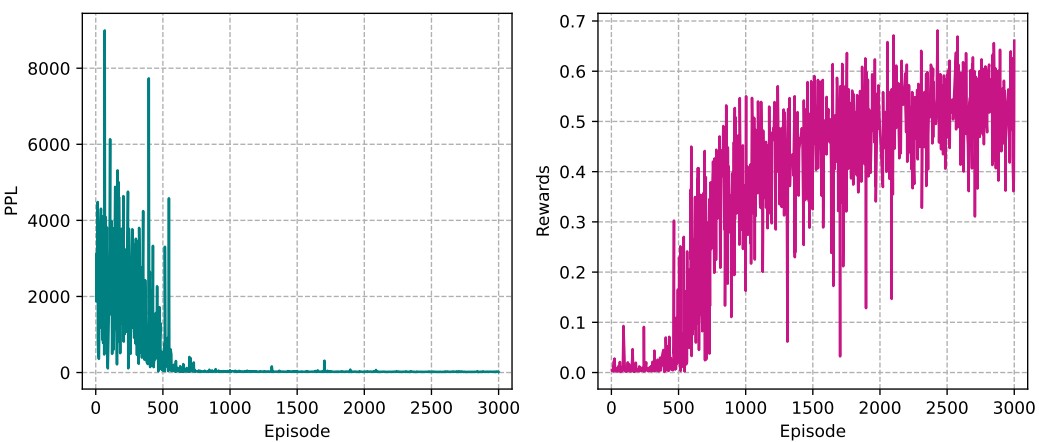

Figure 10: Reinforcement learning reward curve and perplexity curve for OPT-2.7B under structured pruning with a sparsity ratio of 20% on Wikitext.

## C  SEARCHED STRATEGY

Fig. 11 depicts the pruning strategies obtained from specific models in the OPT and LLaMA families, under structured pruning in the environment and agents we designed. Fig. 12 presents a comparison between the sparsity strategy we searched for and that of OWL within the scope of unstructured pruning.

Fig. 13 shows the searched strategies for the intermediate episodes during a search process of 3000 episodes. As can be seen, the sparsity ratio presents a somewhat random distribution without a specific trend in the initial part of the search (the $0$-$th$ episode). In the later stages of the search process, there is considerable overlap in the strategies of the $1999$-$th$ and $2999$-$th$ episodes, indicating that the sparsity strategy is gradually stabilizing.

## D  INFERENCE SPEED

We measured the inference time of the model after pruning with our method. Figure 14 shows the inference time of the OPT-6.7B after 20% sparsity pruning on two 4090 GPUs. It can be seen that our approach reduces the inference time of the model. Compared with the 20%-SliceGPT that introduces extra parameters, our method has a better speedup.

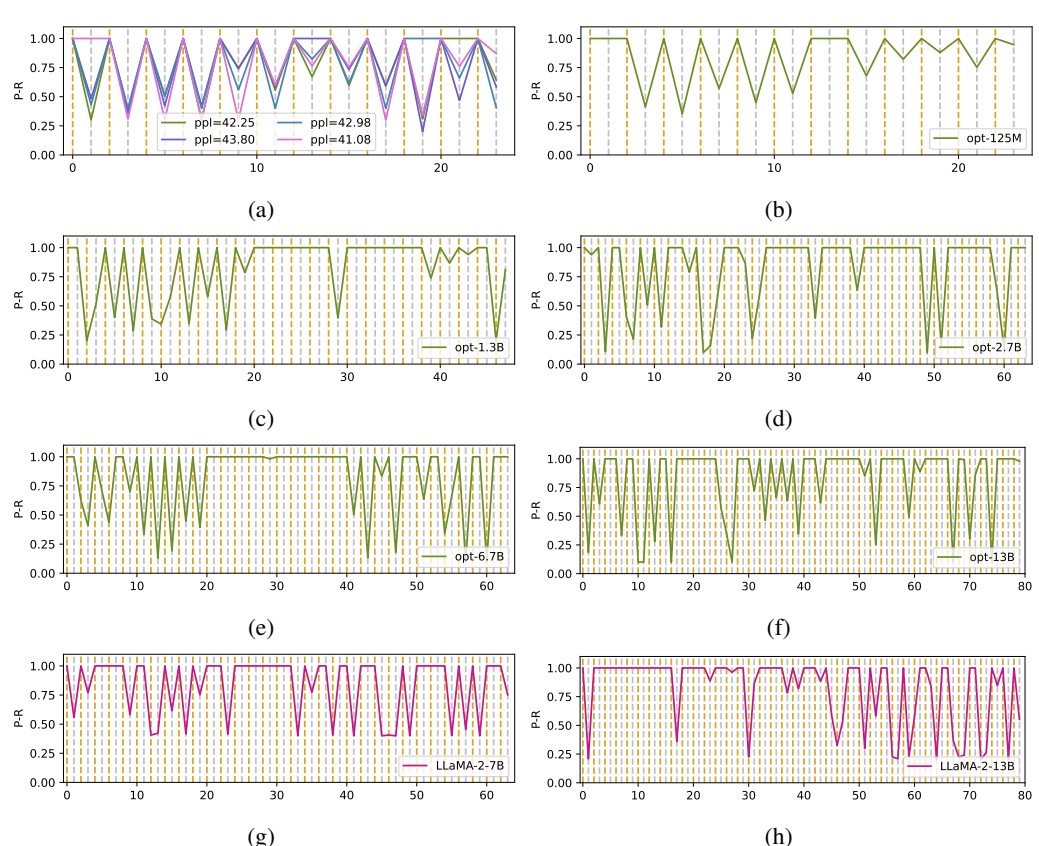

Figure 11: The searched pruning strategies of OPT-125M/1.3B/2.7B/6.7B and LLaMA-V2-7B/13B on WikiText. The horizontal axis is the index of the layer, while the vertical axis "P-R" represents the preserved ratio of parameters in each layer. The yellow dashed line represents the MHA layer, and the gray dashed line represents the FFN layer. (a): A comparison of four different searched strategies for OPT-125M at a sparsity of 30%. (b)-(f): The searched pruning strategies of OPT-125M/1.3B/2.7B/6.7B/13B and LLaMA-V2-7B/13B at a sparsity of 20%.

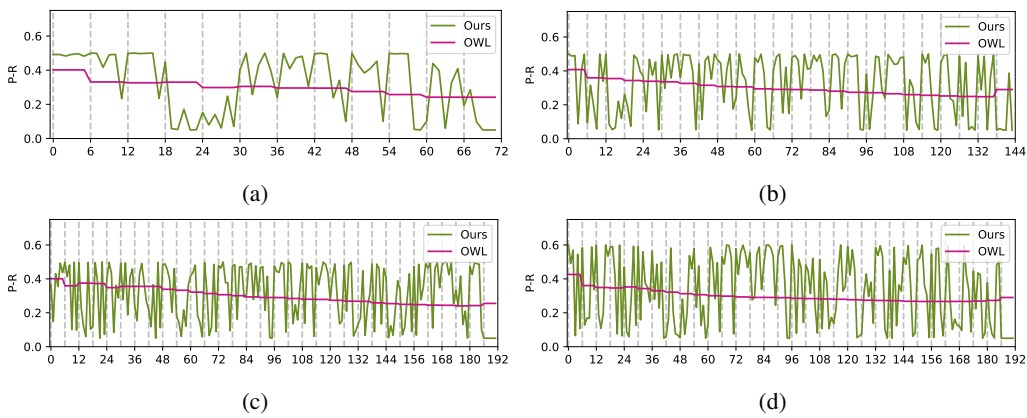

Figure 12: The searched unstructured pruning strategies of OPT-125M/1.3B/2.7B/6.7B on WikiText at a sparsity of 70%. The horizontal axis is the index of the weight, while the vertical axis "P-R" represents the preserved ratio of parameters in each weight. (a): OPT-125M. (b): OPT-1.3B. (c): OPT-2.7B. (d): OPT-6.7B.

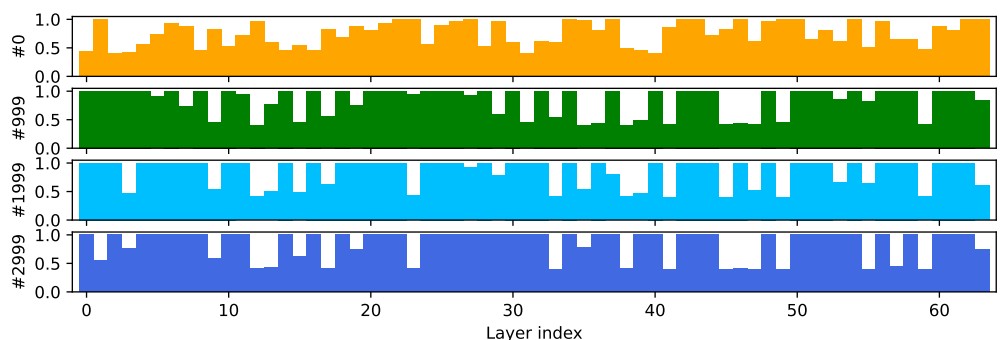

Figure 13: The pruning strategies obtained from the intermediate episodes during the search process for LLaMA-V2-7B at a sparsity of 20%.

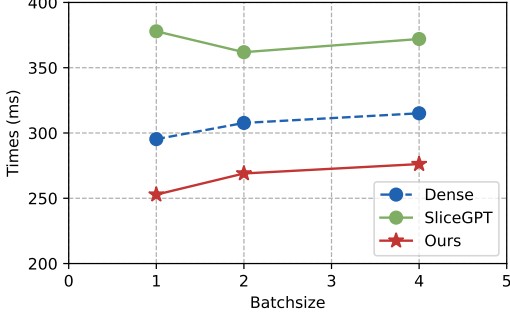

Figure 14: Comparison of inference speed of the OPT-6.7B.

