# OpenReview forum: "SparsitySolver: Efficient Reinforcement Learning-based Pruning for LLMs"
_ICLR.cc/2025/Conference — ICLR 2025 Conference Withdrawn Submission_

### Official Review · Reviewer_st86 · 2024-10-30

**Soundness:** 1
**Presentation:** 2
**Contribution:** 2
**Rating:** 3
**Confidence:** 4

**Summary:**

This paper adapts the RL-based pruning framework for LLMs which has been previously shown to be effective in CNNs but performs poorly when applied to LLMs. It achieves it by densifying the sparse reward of the previous environment for the RL algorithm to efficiently utilize. Additionally, it introduces Compensation Through Reconstruction to alleviate the effect of the structured pruning. Lastly, experiments are conducted on popular LLMs like LLama V1-2 and OPT and results are competitive with other popular pruning methods.

**Strengths:**

* Paper is well written and easy to follow
* Its novelty is redesigning the reward function, which shows it significantly impacts performance, unlike the previous approach.
* Proposed methodology supports both structured and unstructured pruning
* It does not introduce additional parameters for structured pruning
* Results are strong in terms of both perplexity and downstream task performance.

**Weaknesses:**

* The main issue with this paper is its evaluation setup and the lack of comprehensive experiments compared to previous pruning strategies.

* The reason why its evaluation setup is biased is because of the following: unlike previous pruning strategies, this paper defines a proxy reward in terms of perplexity which at first glance makes quite a sense. However, perplexity does not always translate to better performance(i.e. FLAP vs wanda-sp from FLAP paper). Nevertheless, perplexity is still a valid metric for many papers but not for this one because pruning is determined by minimizing the perplexity of a certain dataset which is wikitext eval in this case.  Unsurprisingly, this strategy performs quite good perplexity but I think this leads to reward hacking (Skalse et al., 2022). OPT-6.7B is a good example of this phenomenon where the pruned model has better perplexity than the dense. To settle all these potential issues, I think downstream evaluation should be prioritized because of the sensitivity of the perplexity. However, throughout the paper perplexity is the main metric whereas downstream evaluation is only performed for LLama-V1 7B with only 20 percent sparsity for structured and 70 percent for unstructured pruning. Thus, downstream evaluation should be performed for different models, model sizes, and sparsity ratios. Lastly, the perplexity of a different dataset also can be reported instead of training one.i.e.  C4, PTB.


* Moreover, some results are missing. For example, OWL with SparseGPT has a better perplexity score than the proposed methodology at 70 percent sparsity and is also as good as at downstream evaluation(SparseGPT: 48.03 vs 48.11 of the paper), which shows that the proposed methodology is not SOTA. One more note: even though I appreciate the reproduction, there are some discrepancies between reproduced scores and reported metrics from the original paper, i.e. BESA.  So I believe it is a good idea to include one from the paper and in addition your reproduction results.


References
Skalse, J., Howe, N.H., Krasheninnikov, D., & Krueger, D. (2022). Defining and Characterizing Reward Hacking. ArXiv, abs/2209.13085.

**Questions:**

Q1) What is the overall runtime against other pruning strategies?

Q2) In Figure 1 reward is defined as $1/ppl$ whereas it is $10/ppl$ in the reward function which one is the correct one? If it is $10/ppl$, why do you scale the reward, does it provide any benefits?

---

### Official Review · Reviewer_pUo8 · 2024-11-02

**Soundness:** 2
**Presentation:** 1
**Contribution:** 1
**Rating:** 3
**Confidence:** 5

**Summary:**

This paper presents SparsitySolver, a reinforcement learning-based approach for pruning Large Language Models (LLMs). The method consists of two main technical components: a reinforcement learning framework that searches for optimal pruning strategies, and a reconstruction compensation method for recovering performance in structured pruning without introducing additional parameters. The authors propose a simplified reinforcement learning environment design where the state is represented by the total pruning ratio and the action determines layer-wise sparsity. They also introduce a parameter-free reconstruction compensation method that aims to restore the performance of structured-pruned models through linear combinations of preserved channels. The approach is evaluated on various LLM architectures (OPT, LLaMA-V1/V2, Mistral) across different model scales and pruning granularities (both structured and unstructured), with experiments on language modeling perplexity and zero-shot tasks. The method claims to achieve competitive performance compared to state-of-the-art pruning approaches while maintaining efficiency in the pruning strategy search process.

**Strengths:**

1. The paper proposes a novel direction by incorporating reinforcement learning into LLM pruning strategy search. The idea of using RL to automate pruning strategy discovery shows some originality.
2. The reconstruction compensation method for structured pruning is interesting, as it aims to restore performance without introducing additional parameters, which could be practically valuable.
3. The method demonstrates some level of versatility by supporting both structured and unstructured pruning across different model scales.

**Weaknesses:**

1. **Limited Technical Novelty and Theoretical Foundation:**
   1.1. The motivation for using RL in pruning strategy search is insufficiently justified. The paper fails to establish why RL is particularly suitable for this task compared to other approaches.
   1.2. The RL environment design appears overly simplistic. Using only the total pruning ratio as the state space lacks theoretical justification and seems naive.
   1.3. The derivation of the reconstruction compensation method lacks mathematical rigor. Many assumptions and steps are not properly justified or explained.

2. **Significant Experimental Deficiencies:**
  2.1. The experimental evaluation lacks comprehensive comparisons with other RL-based pruning methods in the literature.
  2.2. No statistical significance analysis is provided for the reported results.
  2.3. The zero-shot evaluation is superficial and fails to demonstrate the method's advantages conclusively.
  2.4. The ablation studies are incomplete and fail to validate the necessity of each component.

3. **Poor Paper Presentation:**
  3.1. The overall organization is chaotic with inconsistent paragraph spacing and formatting.
  3.2. The figures, especially Figure 1, are of low quality and fail to effectively illustrate the proposed method.
  3.3. Mathematical notations and equations lack proper explanations and context.
  3.4. Citations are inconsistently formatted and poorly integrated into the text.

4. **Insufficient Technical Details:**
 4.1. Critical implementation details of the RL agent architecture and training process are missing.
 4.2. The practical aspects of the reconstruction compensation method are unclear.
 4.3. The computational overhead and deployment considerations are not adequately discussed.

**Questions:**

1. **Methodology:**
  1.1. What is the justification for such a simplified RL state space design? Have you considered incorporating model structural information into the state representation?
  1.2. How does the reconstruction compensation method handle different types of network layers? What guarantees its effectiveness?

2. **Experiments:**
  2.1. Can you provide detailed comparative experiments with existing RL-based pruning methods?
  2.2. What are the results on larger-scale models (>70B parameters)?
  2.3. Have you conducted stability analysis across different downstream tasks?

3. **Practical Implementation:**
 3.1. What is the end-to-end training time and computational resource requirement?
 3.2. How do you balance the search cost versus performance improvement in practical deployments?
 3.3. Are there any architectural limitations to your method's applicability?

---

### Official Review · Reviewer_CGeK · 2024-11-03

**Soundness:** 3
**Presentation:** 3
**Contribution:** 3
**Rating:** 6
**Confidence:** 3

**Summary:**

This paper addresses the shortcomings of current pruning methods that overlook the inner sparsity differences between layers in large language models (LLMs). The motivation is clear and straightforward, and the design of SparsitySolver integrates reinforcement learning to facilitate an intuitive exploration of various pruning granularities. Additionally, the implementation of performance compensation through a linear combination in the final linear layer is both simple and effective. Experiments demonstrate its effectiveness across both general perplexity (PPL) and LLM benchmarks.

**Strengths:**

1. The illustrations for this work are easy to understand, and the method's description is clear and straightforward.
2. The approach demonstrates its effectiveness across multiple large language models (LLMs) of varying scales when compared to current state-of-the-art (SOTA) methods. Moreover, the pruned performance shows improvements in both general metrics, such as perplexity (PPL), and current LLM benchmarks.
3. Additionally, the proposed search strategy highlights the diverse sparsity within LLMs across layers and reflects the overall sparsity distribution within these models, presenting an interesting observation for further research.

**Weaknesses:**

1. To gain a comprehensive understanding of a pruning technique, it is essential to evaluate its performance on complex tasks such as machine translation and other capabilities of large language models (LLMs), including in-context learning (few-shot).
2. One pressing application of pruning techniques is to reduce the operational costs of very large-scale language models, such as LLaMA2-70B and PaLM-540B. However, this paper does not provide support for these models.

**Questions:**

1. Quantization is a direct approach to reducing storage costs and inference time. Is this method compatible with existing quantization techniques?

---

### Official Review · Reviewer_Jubx · 2024-11-04

**Soundness:** 2
**Presentation:** 1
**Contribution:** 2
**Rating:** 3
**Confidence:** 4

**Summary:**

The authors introduce RL into LLM pruning. I feel the paper is tough to understand as the authors always give some reverse statements.

**Strengths:**

Introduce RL into LLM pruning.

**Weaknesses:**

Bad Writing (why reinforcement learning is a good choice? Why you say the problem of previous research and say your work is the first one.)

"Suchmanuallyorsemi-manuallydesignedprun
ingstrategiesoftenleadtosuboptimalresults,whichmakesreinforcementlearn
ingafeasiblesolution. However, current reinforcement learning-basedpruning
 methodsusuallyhaveredundantenvironmentdesignsormultipleagents, render
ingthemill-suitedtomassiveLLMs. Hence,weproposeSparsitySolver,which
 first incorporates reinforcement learningintothepruningofLLMs, supporting
 variouspruninggranularity."

**Questions:**

RL is known unstable. How to reverse the parameters that you have pruned before?
Does this method require training or just forward pruning?
For the reward compensation, will that increase reduce sparsity as the sparse item for different channels are different.
What is the meaning of without need for additional computation?
 "Weproposea simpleandefficient reinforcement learningenvironment, improving the
 sparserewardenvironmentinexistingRLpruningmethodswithouttheneedforadditional
 computation."

---

### Note · Authors · 2024-11-18

I have read and agree with the venue's withdrawal policy on behalf of myself and my co-authors.